# Transcriptomic, Hormonomic and Metabolomic Analyses Highlighted the Common Modules Related to Photosynthesis, Sugar Metabolism and Cell Division in Parthenocarpic Tomato Fruits during Early Fruit Set

**DOI:** 10.3390/cells11091420

**Published:** 2022-04-22

**Authors:** Miyako Kusano, Kanjana Worarad, Atsushi Fukushima, Ken Kamiya, Yuka Mitani, Yozo Okazaki, Yasuhiro Higashi, Ryo Nakabayashi, Makoto Kobayashi, Tetsuya Mori, Tomoko Nishizawa, Yumiko Takebayashi, Mikiko Kojima, Hitoshi Sakakibara, Kazuki Saito, Shuhei Hao, Yoshihito Shinozaki, Yoshihiro Okabe, Junji Kimbara, Tohru Ariizumi, Hiroshi Ezura

**Affiliations:** 1Faculty of Life and Environmental Sciences, University of Tsukuba, 1-1-1 Tennodai, Tsukuba 305-8572, Japan; worarad.kanjana.ga@u.tsukuba.ac.jp (K.W.); yo.shinozaki92@gmail.com (Y.S.); y-okabe@nippn.co.jp (Y.O.); ariizumi.toru.ge@u.tsukuba.ac.jp (T.A.); ezura.hiroshi.fa@u.tsukuba.ac.jp (H.E.); 2Tsukuba-Plant Innovation Research Center (T-PIRC), University of Tsukuba, 1-1-1 Tennodai, Tsukuba 305-8572, Japan; 3RIKEN Center for Sustainable Resource Science, Yokohama 230-0045, Japan; afukushima@kpu.ac.jp (A.F.); indi5blue@ezweb.ne.jp (Y.M.); yozo.okazaki@bio.mie-u.ac.jp (Y.O.); yasuhirohigashi01@gmail.com (Y.H.); roy.nakbayaski@gmail.com (R.N.); kobamako@riken.jp (M.K.); tetsuya.mori@riken.jp (T.M.); tomoko.nishizawa@riken.jp (T.N.); yumiko.takebayashi@riken.jp (Y.T.); mikiko@riken.jp (M.K.); sakaki@agr.nagoya-u.ac.jp (H.S.); kazuki.saito@riken.jp (K.S.); 4Graduate School of Life and Environmental Sciences, Kyoto Prefectural University, 1-5 Shimogamohangi-cho, Sakyo-ku, Kyoto 606-8522, Japan; 5Graduate School of Life and Environmental Science, University of Tsukuba, 1-1-1 Tennodai, Tsukuba 305-8572, Japan; k.kamiya1115@gmail.com (K.K.); haoshu1124@gmail.com (S.H.); 6Graduate School and Faculty of Bioresources, Mie University, Tsu 514-8507, Japan; 7Graduate School of Bioagricultural Sciences, Nagoya University, Nagoya 464-8601, Japan; 8Rijk Zwaan Breeding Japan K.K., Chiba 289-1608, Japan; j.kimbara@rijkzwaan.com; 9Research Institute, Kagome Co., Ltd., 17 Nishitomiyama, Nasushiobara 329-2762, Japan

**Keywords:** tomato, parthenocarpy, metabolomics, transcriptomics, next-generation sequencing, correlation network analysis

## Abstract

Parthenocarpy, the pollination-independent fruit set, can raise the productivity of the fruit set even under adverse factors during the reproductive phase. The application of plant hormones stimulates parthenocarpy, but artificial hormones incur extra financial and labour costs to farmers and can induce the formation of deformed fruit. This study examines the performance of parthenocarpic mutants having no transcription factors of *SlIAA9* and *SlTAP3* and *sldella* that do not have the protein-coding gene, *SlDELLA*, in tomato (cv. Micro-Tom). At 0 day after the flowering (DAF) stage and DAFs after pollination, the *sliaa9* mutant demonstrated increased pistil development compared to the other two mutants and wild type (WT). In contrast to WT and the other mutants, the *sliaa9* mutant with pollination efficiently stimulated the build-up of auxin and GAs after flowering. Alterations in both transcript and metabolite profiles existed for WT with and without pollination, while the three mutants without pollination demonstrated the comparable metabolomic status of pollinated WT. Network analysis showed key modules linked to photosynthesis, sugar metabolism and cell proliferation. Equivalent modules were noticed in the famous parthenocarpic cultivars ‘Severianin’, particularly for emasculated samples. Our discovery indicates that controlling the genes and metabolites proffers future breeding policies for tomatoes.

## 1. Introduction

Tomatoes (*Solanum lycopersicum*) are one of the main crops cultivated in the world. Since improving fruiting productivity can enhance the yield of tomatoes, the investigation into the mechanism of fruiting and subsequent fertilisation of tomatoes has been studied thoroughly. Parthenocarpy is the cultivation of seedless fruit, which is not a function of pollination but can improve fruit set when the conditions are not favourable during the reproductive phase. Therefore, parthenocarpy can improve winter and early yields, enabling consumers to purchase crops all year round [1,2]. Additionally, the food industry prefers seedless fruits due to their longer shelf life and ease of processing.

Parthenocarpy can be facilitated artificially by applying several plant hormones [3,4]. However, the use of artificial plant hormones incurs extra financial and labour costs to farmers and leads to small-size cultivation and the development of malformed fruit with poor firmness [5,6,7]. Additionally, many of the current parthenocarpic mutants have a substantial impact on the entire plant’s morphogenesis, such as leaf morphological changes, the development of malformed fruits with reduced fruit size and quality and a reduced fruiting rate [8,9,10]. Thus, it is crucial to investigate the significant molecular targets that can achieve parthenocarpy but do not affect morphological alterations of other organs and characteristics of the entire plant.

Genes associated with plant hormones are presently being used to produce parthenocarpic mutants in tomatoes. Plant hormones play a critical role in fruit development, and plant hormones are recognized to be closely related to parthenocarpy [11]. A model in which auxin and gibberellin (GA) regulatory pathways cooperate hierarchically has been proposed. Pollination initiates the auxin regulatory pathway; after that, the GA-regulatory pathway is activated [12]. Therefore, to avoid negative impacts and develop seedless tomatoes with better yield and quality through molecular breeding and transgenic approaches, it is necessary to ascertain the molecular mechanisms of parthenocarpy via plant hormone-mediated control. Auxin and GA are the primary hormones that encourage fruit initiation and stimulate growth to enhance fruit production [13,14]. Two prominent mutants that demonstrate parthenocarpy include *aux/indole-3-acetic acid 9* (*sliaa9*) and *sldella*. *Sliaa9* works as a transcription factor (TF) in the control of the expression of auxin-responsive factors (ARFs) through the auxin signalling that is involved in the regulating the fruit set [15,16]. Wang and colleagues found that the downregulation of *SlIAA9* in tomatoes caused parthenocarpy [16]. In contrast, *SLDELLA* is a negative regulator of GA signalling by combining with the GA receptor, GA-INSENSITIVE DWARF1 (GID1) [17]. The loss of *SLDELLA* function in the GA signalling pathway leads to *SLDELLA*, which generally inhibits GA-responsive gene expression by binding to the corresponding TFs. In the presence of GA, post-translational modification and degradation of *SLDELLA* release the suppression of GA-responsive gene expression [14,17,18]. In tomatoes, mutations and RNAi inhibition of *SLDELLA/PROCERA* cause parthenocarpy [18,19,20], confirming that *PROCERA* is a repressor of fruit initiation. Although auxin and GA play an essential role in fruit initiation, the exact mechanism by which auxin and GA promote fruit initiation and the nature of the genes that control fruit set remains unclear. Meanwhile, other genes can cause parthenocarpy in plants. For example, proteins encoded by MADS-box genes (ABC model) are essential TF families linked with flowering time assessment, fruit development and seed set [21,22,23,24]. Earlier reports have considered the role of the MADS-box genes in parthenocarpy, such as the downregulation of *TM29*, a homologue of *SEPALLATA*, which causes the changing of both petals and stamens into sepals followed by pistil infertility, leading to parthenocarpic fruits in tomatoes [25]. Similarly, the loss of function of *APETALA3* (*TAP3*), a B class of a floral organ identity gene that converted into a carpelloid structure, reveals parthenocarpic fruits in tomatoes [26,27]. In parthenocarpic cultivars, the Russian cultivar ‘Severianin’ is a natural source of parthenocarpy in tomatoes, possessing the *pat-2* gene located on chromosome 4. This cultivar demonstrates strong parthenocarpy even under several environmental conditions [28]. Therefore, many investigations have been done to ascertain the characterisation of *pat-2* in ‘Severianin’ [29,30,31].

Multi-omics strategies allow us to derive new insights into understanding complicated biological mechanisms in plants. Of these, data integration of transcriptomic and metabolomic data can be applied to measure plant responses, e.g., biotic and abiotic stresses, senesce, chemical treatments including plant hormones and the mutation of significant genes [32,33,34,35]. A combination of varying separation techniques hyphened with mass spectrometry is applied to analyse metabolites in central metabolism, specialised metabolites, lipids and plant hormones for covering a broad range of metabolites [36,37,38,39].

Correlation network analysis helps visualize and give hints for interpreting the biological relationships between multiple variables (nodes). Furthermore, constructing transcript and metabolite correlation networks enables extracting major modules that are necessary for natural events. For example, the correlation network analyses were highlighted to reorganize changes of metabolite and transcript status in *Arabidopsis thaliana* (Arabidopsis), tomato, pepper, potato and *Brassica rapa* [40,41,42,43].

In this research, we sought to examine the molecular events from the varying behaviour of parthenocarpic mutants, i.e., *sliaa9*, *sldella* and *sltap3*, using multi-omics strategies to interpret the techniques involved in early fruit growth. First, we discussed the plausible roles of plant hormones verified by hormonome assessment in the regulation of fruit set and the timing of their changes. We then revealed the molecular activities causing fruit set process via the joint transcriptomic and metabolomic approach. Additionally, to better explain the intersection between each mutant and wild type (WT) during fruit setting and formation in tomatoes, we adopted RNA-sequencing (RNA-seq) to profile transcript alterations involved in fruit set in the prominent parthenocarpic cultivar ‘Severianin’. Our investigation revealed that the genotype-dependent behaviour appears vital in controlling the expression of transcripts, especially on the flowering day. Moreover, the comparative examination at the transcript and metabolite concentrations of pollination-induced and parthenocarpic fruit sets identified auxin and GA, photosynthesis and cell division and glycolysis as prominent actors of the fruit sets.

## 2. Materials and Methods

### 2.1. Plant Materials and Growth Conditions

In this study, Japanese (MTJ) and Brazilian (MTB) varieties of Micro-Tom were used as a wild type of tomato (*Solanum lycopersicum*), and *sliaa9* with MTJ background, *sldella* with MTB background and *sltap3* with MTJ background were used as mutants generated by ethyl methanesulfonic acid (EMS) treatment. Seeds were absorbed overnight in deionized water and stored at 25 °C for 4 to 6 days on deionized water-moistened filter paper under a photoperiod of 16 h/8 h with a light intensity of 100 µmol·m^−2^·s^−1^. Tomato seedlings were transplanted into Rockwool cubes (75 mm × 75 mm × 65 mm) and grown in a nutrient solution with an electrical conductivity (EC) of 1.6 dS·m^−1^ (Otsuka A, Otsuka Chemical Co., Ltd., Osaka, Japan) under fluorescent light with a photoperiod of 16 h/8 h and a light intensity of 300 µmol·m^−2^·s^−1^ under light conditions of 25 °C in bright conditions and 20 °C in dark conditions. The experimental treatment was divided into two wards: one was manually pollinated on the day of flowering, and another was manually emasculated. Day after flowering (DAF) was used as a reference for the sampling of pistils from each line at two days before flowering (–2DAF, flowering day (0DAF), 2 days after flowering (2DAF), 4 days after flowering (4DAF) for the three mutants with corresponding WT and 8 days after flowering (8DAF) for the *sltap3* mutant and MTJ. Sampled ovaries were weighed and subsequently frozen at −80 °C in liquid nitrogen until use. The ‘Severianin’ and ‘M82’ cultivars were grown in nutrient culture in a coconut–peat substrate fed with hydroponic solution (revised-A nutrient prescription, Otsuka Chemicals, Osaka, Japan) with an EC of 0.8–1.5 ms·cm^−1^. Appendix A shows the experimental design.

### 2.2. Microarray Analysis

For the quality check of isolated RNA, the Agilent RNA6000 Nano LabChip^®^ kit was used for 1 μL of 250 ng/3 μL RNA. According to the kit protocol, the quality assessment was performed using an Agilent 2100 Bioanalyzer. According to the kit protocol, the labeled cRNA was then purified for the same concentrations of RNA using the Gene Chip^®^ 3′ IVT Express Kit (Affymetrix). The content was obtained using a NanoDrop and was sufficient. According to the kit protocol, the labelled cRNA was prepared to 15 μg/32 μL and fragmented. After that, 80 μL of fragmented and labelled cRNA and hybridisation cocktail mixture was injected into the GeneChip^®^ Tomato Genome Array and incubated at 45 °C at 60 rpm for 16 h according to the protocol. Next, the microarray chips of the microarray were set into the GeneChip^®^ Fluidics Station 450 (Affymetrix), and the chips were washed and stained according to the accompanying manual. Finally, the GeneChip scanner 3000 (Affymetrix) was utilised to scan the chip and measure the gene expression level. Our microarray data have deposited to NCBI GEO.

### 2.3. Metabolomic Profiling Analysis

The processing and extraction of freeze-dried samples in the 2 mL tubes were mixed thoroughly with 5 mm of zirconia beads in a mixer mill (MM301; Retsch, Haan, Germany) for 1 min at 20 Hz. After mixing, the samples were used for gas chromatography hyphened with time-of-flight MS (GC-MS) and liquid chromatography hyphened with quadrupole time-of-flight mass spectrometry (LC-MS) for polar metabolites and lipids. Hormonome analysis was utilized by LC with a tandem quadrupole mass spectrometer (LC-q-MS/MS). Additionally, carotenoid and chlorophyll content was estimated using liquid chromatography hyphened with photodiode array (LC-PDA). We used 5–6 biological replicates for metabolomic analysis and six carotenoid quantification, while three biological replicates were used to quantify 42 hormones. For detailed information, see Appendix A.

### 2.4. RNA-Seq Analysis

Ovaries of Severianin and M82 (100 mg fresh weight (FW)) were harvested in the greenhouse located at Nasushiobara, Tochigi, Japan. Moreover, we ordered genome sequencing and RNA-seq analysis to BGI JAPAN (the Beijing Genomic Institute, Kobe, Japan). Illumina HiSeq 2000 (Illumina) was used for the RNA-seq analysis. For detailed information, see Appendix A.

### 2.5. Weighted Correlation Network Analysis (WCNA) of Transcriptome and Metabolome Datasets

WCNA was carried out using transcriptome and metabolome datasets. In more detail, see Appendix A.

### 2.6. Data Analysis

The following analyses were carried out mainly using R (ver. 3.6.3), a statistical analytical tool. For the detail information, see Appendix A.

## 3. Results

### 3.1. Comparison of Ovary/Fruit Development of WT and the Mutants during Early Fruit Set

We weighed ovary samples of the three mutants, *sliaa9*, *sldella* and *sltap3*, and the corresponding WT (MTJi, MTB and MTJt) at developmental phases during fruit set: before flowering (–2DAF), during flowering (0DAF), 2 and 4 days after flowering (2DAF, 4DAF) and 8 days after flowering for the *sltap3* mutant (8DAF) with and without pollination (Appendix A). The fruit enlargement of the *sltap3* mutant at 8DAF was comparable with that of WT and the two mutants at 4DAF. For each experiment, transcriptomic, metabolomic and plant hormone alternations were linked with all categories of fruit set (Appendix A). Regarding phenotypes of parthenocarpic mutants related to *SlIAA9*, *SlDELLA* and *SlTAP3* from previous research, the three mutants demonstrated unusual morphological phenotypes even though these mutants have a high proportion of parthenocarpic fruit set (Appendix A). At the 0DAF stage, the *sliaa9* mutant exhibited increased pistil growth compared with the other two mutants and the WT. After the 0DAF, the FW of each mutant was significantly higher than WT without pollination. Notably, the new weight of the *sliaa9* mutant was already high before pollination (Figure 1). In the case of the *sldella* mutant, there was a significant alternation in weight with or without pollination at 4DAF. The emasculated *sltap3* mutant influenced the fruit enlargement at the 8DAF stage, similar to the emasculated *sldella* mutant at the 4DAF stage. To ascertain the molecular mechanism of parthenocarpic fruit productivity in floral homeotic mutants such as the *sltap3*, the *sltap3* mutant samples were utilised until the 8DAF stage.

### 3.2. Plant Hormone Accumulation Patterns during Fruit Set in WT and the Mutants with and without Pollination

To examin the plant hormone alternations during the fruit set, hormonome analysis was conducted for the three mutants and corresponding WTs at varying stages with and without pollination (Figure 2 and Appendix A). In the 42 hormones (indole acetic acid (IAA) and their derivatives, GAs, cytokinins (CKs), abscisic acid (ABA), salicylic acid (SA) and jasmonic acid (JA) that were assessed throughout the mutants and corresponding WTs (Appendix A), the accumulative composition of the five hormones in the *sl**iaa9* mutant varied from that of the *sldella* and *sltap3* mutants (Figure 2). In contrast to WT and the other mutants, the *sliaa9* mutant pollination efficiently stimulated the increase in auxin (IAA) after flowering; the maximum was about 3000 pmol/g FW. The levels of bioactive GAs (GA1, GA4 and GA20) were also significantly induced (about 100 times) by *sliaa9* pollination compared with others. Additionally, the accumulation configurations of *trans*-zeatin content (tZ) in the mutants and WTs differed (Figure 2 and Appendix A).

### 3.3. General Trend of the Transcriptomic and Metabolomic Changes of the Mutants

Microarray analysis, metabolite profiling and carotenoid and chlorophyll quantification were performed to capture complete transcriptomic and metabolomic alterations before and after anthesis of tomato fruits (Appendix A). The 715 metabolites, including primary metabolites, polar secondary metabolites, lipids, carotenoids and chlorophylls, were identified or annotated in the metabolomic data (Appendix A). Principal component analysis (PCA) was conducted using transcript and metabolite profiles to visualize the sample distribution (Figure 3). The profiles of the *sltap3* mutant lack pollination data due to the pollen fertility of the mutant. The PC1 direction throughout the three mutants and WTs explained the separation in the PCA according to the developing fruit status of the mutants and WTs and WT without pollination concerning transcript and metabolite profile data, respectively (Figure 3).

The transcript profiles of the *sliaa9* mutant at the 0DAF stage (iaa9 0) were similar to those before flowering (iaa9 m2) and WT before flowering (MT-J m2), although the ovary weight of the *sliaa9* mutant was higher than that of the iaa9 m2 and MT-J m2 samples. After flowering, the *sliaa9* mutant and WT profiles had advanced sequentially. The *sldella* mutant demonstrated fundamental expression configurations compared with the result of the *sliaa9* mutant profiles. After flowering, the expression configurations of the *sldella* mutant with and without pollination showed the same pattern with WT pollination (MT-B 2P, MT-B 4P). Simultaneously, WT unpollination (MT-B 2E, MT-B 4E) was separated. In the case of the *sltap3* mutant, time course-dependent distinction was observed from the 0DAF stage according to fruit development.

Metabolite profiles of the *sliaa9* and *sldella* mutants at the 0DAF (iaa9 0 and della 0) were distinguished clearly from other samples. In contrast, the *sltap3* samples at the 0DAF stage (tap3 0) were gathered with the profiles of the *sltap3* mutant at -2DAF, 2DAF and 4DAF together with WT at 2DAF. Samples of the *sltap3* mutant at the 8DAF stage (tap3 8) and those of WT with pollination at 4DAF (MT-J 4P) were close to each other on the PCA score scatter plot. The outcomes are uniform, with similar pistil weight for WT and the *sltap3* mutant (Figure 1).

### 3.4. WCNA of the Transcript and Metabolite Profiles of the Mutants

To understand the responsive network structure in each tomato mutant, an unsigned correlation network linkage was designed using the transcript and metabolite profile data with the fast-greedy modularity optimisation algorithm (Appendix A). As a result, the networks had five significant modules in the transcript and metabolite networks (Figure 4 and Figure 5).

Gene ontology (GO) enrichment assays using genes from each module demonstrated that the core network was composed of developmental genes related to the photosynthesis, cell cycle, protein kinase complex and oxidoreductase activity (Figure 4A; Appendix A). Photosynthesis/chloroplast was a standard module throughout the three mutants. Among them, significantly overrepresented genes were photosynthesis-linked genes in the *sliaa9* mutant network (M3). In this module, the ‘hub’ genes associated with many other nodes in M3, including *glucose-1-phosphate adenylyltransferase* (77 edges), *ribulose bisphosphate carboxylase small chain 1* (73 edges), *fructose-1,6-bisphosphatase 1* (71 edges), *photosystem II 22 kDa protein* (70 edges), etc. (Figure 4B; Appendix A). In the *sldella* mutant network, significantly overrepresented genes were found in all four modules, particularly the cell cycle in M2 (Figure 4A). This model demonstrated a highly interrelated hub gene of *SYP111* (46 edges), *cytoskeletal END BINDING PROTEIN 1C* (EB1C; 43 edges), mitotic spindle checkpoint protein *MAD2* (38 edges), etc. We found many genes related to cell division and expansion upregulation, such as *cyclin A1, B2-type cyclin* and *cyclin B2* (Figure 4B). Interestingly, genes such as *GA20-ox1* (37 edges) and *ethylene response factor 1* (63 edges) were related to plant hormones and cell division and expansion, and genes such as auxin responsive protein-coding genes (43 edges) for hormone-related genes were related to auxin signalling, while *cycD3* (36 edges) and *expansin* (33 edges) appeared as the cell division and expansion. These also participate in the M2 module in the *sldella* network. The GO term of ‘oxidoreductase activity’ in M1 was significantly overrepresented in the *sltap3* mutant linkage including a highly interrelated hub gene (Figure 4A). For instance, probable *xyloglucan endotransglucosylase* (182 edges) demonstrated the most remarkable association with another module, followed by *4-hydroxyphenylpyruvate dioxygenase* (176 edges), *beta-galactosidase* (171 edges), *glucose-1-phosphate adenylyltransferase* (159 edges), etc. Furthermore, M1 in the *sltap3* network contained *gibberellin 20-oxidase-1* and *expansin*, essential genes in cell proliferation and division (Figure 4B).

In the metabolite correlation linkages of the *sliaa9* and *sldella* mutant profiles, the number of metabolites in each module was less than that of the *sltap3* mutant (Figure 5A; Appendix A). The four hub metabolites (phenylalanine, tocopherol, sitosteryl linoleate and ketopantoate) found in the *sliaa9* mutant linkage in M3 can be synthesized in plastid (Figure 5B). Glucuronic acid, which is the common precursor for arabinose, xylose, galacturonic acid and apiose residues found in the cell wall was connected with cell-division-related genes in the *sldella*-M2 module (Figure 5B). In the *sltap3* mutant linkage, galacturonic acid, gluconic acid, glucuronic acid and saccharic acid were isolated as hub metabolites, while proteogenic amino acids were linked with the others (Figure 5B).

### 3.5. Transcript Changes in Photosynthesis, Sugar Metabolism and Cell Wall Biosynthesis during Tomato Fruit Set

WCNA can be used for obtaining clusters (modules) of highly linked genes or metabolites. For example, the network assay using transcriptome data isolates a common module ‘photosynthesis’ throughout the mutant networks as a prospective specimen for parthenocarpy. During the early formation of fruit, photosynthesis itself considerably contributes to the organ’s metabolism and growth. Network analysis of hub genes posited that high expression of common genes related to photosynthesis was involved in *protochlorophyllide oxidoreductase* (*SlNADPH*), *photosystem II 22 kDa protein* (*SlPSBS*) and *tetrapyrrole-binding protein* (*SlGUN4*) in the three mutants and WTs with pollination (Figure 6). This also relates to other photosynthesis-related genes.

The levels of the four glycolysis-related genes were decreased in the unpollinated WTs (Appendix A). In contrast, the *phloem sucrose transporter 1* (*SlSUT1*) and *tomato fructokinase-2* (*SlFRK2*) were abruptly increased in the three mutants and pollinated WTs from the 0DAF stage (Figure 6). The expression level of tomato *sucrose synthase* (*SlTOMSSF*) was also higher in the mutants than the unpollinated WTs (Figure 6). Additionally, the expression of *SlHOMEOBOX 15A* (*SlHB15A*), which was one of the class III homeodomain-leucine zipper gene (HD-ZIP III) transcription factors to mediate crucial developmental processes including fruit set, demonstrated substantial downregulation after flowering except for WTs without pollination (Figure 6) [44,45].

Cell divisions and cell-wall-related genes were upregulated at the post-flowering stage in WTs, while their activation of some genes occurred earlier in the mutants at the flowering stage (Figure 6). Cell-wall-related genes were significantly demonstrated in the mutants, including *expansin 5* (*SlExpa5*) and *endo-1,4-beta-d-glucanase* (*Slcel7*), which participated in fruit enlargement through cell expansion. In the microarray we used for the research, there were 83 genes involving that carbohydrate metabolic process (Appendix A). In differentially expressed genes linked to this process, 12 genes were discovered in each network assay of the mutants. The expression levels of the six genes linked to sugar metabolism and cell wall biosynthesis in the mutants and pollinated WTs were higher than those of WTs without pollination (Appendix A).

### 3.6. Metabolite Correlations in Photosynthesis and Carbohydrate Metabolism and Extracted by Network Analysis

Since the outcome of the transcript profiling indicated the alternations of photosynthesis, sugar metabolism and cell wall biosynthesis, we firstly focused on metabolites related to these metabolic events. As a result, metabolite alternations in photosynthesis and carotenoid pathways in the *sliaa9* and *sldella* mutants and the corresponding WTs demonstrated a comparable trend, while those in the *sltap3* mutant did not (Appendix A).

For sugar metabolism, changes in fructose and glucose showed a clear difference between unpollinated WTs and the others (Figure 7). After flowering, the levels of these sugars in unpollinated WTs were decreased, while the others demonstrated a significant increase. Sucrose is a substrate for the production of fructose and glucose. As expected, the sucrose level of unpollinated WTs was higher than the others. However, this level of unpollinated WTs showed a similar trend, i.e., decreased after flowering (Figure 7). The levels of metabolites produced through glycolysis (e.g., fructose-6-phosphate) remained constant (Appendix A). The levels of metabolites produced through glycolysis were similar with the result of antisense of the IAA9 line described by Wang et al. (2009) [46].

In the levels of metabolites that are crucial for cell wall formation involving the cell wall constituents and lignin biosynthesis, glucuronic acid (cell-wall component), phenylalanine, cinnamic acid and coumaryl alcohol (lignin biosynthesis) demonstrated significant changes (Figure 7). Alternatively, the levels of caffeic acid, ferulic acid and sinapic acid, downstream products from cinnamic acid and flavonoids, including quercetin derivatives, did not exhibit distinct configurations throughout all genotypes (Appendix A). The level of shikimic acid, which is the precursor of aromatic amino acids and phenylpropanoids, used to make pigments, hormones and cell wall constituents, demonstrated a significant increase in the three mutants at the 0DAF stage. However, that of WT was less. This trend was discovered in the changes of phenylalanine (Figure 7).

The common metabolites, phenylalanine, glucuronic acid, 4-hydroxysphinganine and two unidentified glycoalkaloids (UGAs), 6 and 9, were found throughout the three linkages. Of these, 4-hydroxysphinganine was high in the mutants with and without pollination, especially at the 0DAF stage (Figure 7). Similar trends were observed in unidentified glycoalkaloids (UGAs) 6 and 9. Glycoalkaloids have been proposed to function in plant defence against biotic threats (Figure 7). However, tomatine and tomatine derivatives, which were not found as the common metabolites in the three mutant networks, showed no specific patterns (Appendix A).

### 3.7. RNA-Seq Analysis of a Parthenocarpic Cultivar ‘Severianin’ Reveals the Similar Trends of Transcript Changes Found throughout the Three Micro-Tom Mutants

Significant alternations in genes and metabolites linked to photosynthesis, sugar metabolism and cell wall biosynthesis in the three mutants, *sliaa9*, *sldella* and *sltap3*, were discovered. The mutants using a cultivar ‘Micro-Tom’ are expected to show whether parthenocarpic cultivars show similar transcript changes during the fruit set. Thus, an RNA-seq analysis was performed using ‘Severianin’, which demonstrates parthenocarpy by comparing one of the common cultivars, M82. WGCNA was then conducted for genes in the RNA-seq data. The resultant linkages had four modules (Figure 8; Appendix A). Of these, the GO term of M1 (cell wall) was overpresented. The linkage analysis in M1 revealed *beta-galactosidase* (*Solyc10g055470*) as a hub gene. The hub gene is one of the crucial genes for cell wall modification enzymes in various plants, including tomatoes [47]. Additionally, two *pectinesterases* were found in M1 that played multiple functions involved in cell wall metabolism. To envision transcript alternations at each time point, MapMan analysis was performed (Appendix A). There was remarkable upregulation of genes linked to photosynthesis and cell wall in Severianin. This pattern was comparable with transcript alternations in the profiles of the *sliaa9*, *sldella* and *sltap3* mutants, especially for emasculated samples (Appendix A).

## 4. Discussion

In this research, transcript and metabolite profiling were demonstrated to explain the alternations occurring within the growing ovary and fruits and discriminated between the pollination-dependent and pollination-independent fruit sets of WT and parthenocarpy mutants. Furthermore, this examination revealed vast differences in gene expression linked with the formative stages in tomatoes. Additionally, the fundamental reasons were that such studies provide valuable resources for understanding how *sliaa9*, *sldella* and *sltap3* impart their role(s) in gene transcription and subsequently on metabolite accumulation and developmental procedures. Moreover, *Severianin*, a prominent parthenocarpic cultivar, demonstrated a similar linkage structure compared with that of the three parthenocarpic mutants, particularly after emasculation.

### 4.1. Endogenous Accumulation of IAA and Active GAs Has no Relation for Induction of Parthenocarpy of the Mutants Lacking SlIAA9, SlDELLA and SlTAP3

Earlier research discovered that *SlIAA9* and *SlDELLA* are two prominent genes that participate in auxin and gibberellin, which are important for plant growth and development, including fruit set and enlargement through cell division and cell multiplication [16,48,49]. *Sliaa9* demonstrated augmented pistil enlargement compared with the mutants *sldella* and *sltap3* and the WT from the 0DAF stage. Corresponding levels of IAA and bioactive GAs (GA1, GA4 and GA20) in the *sliaa9* mutant with pollination were efficiently stimulated from the 0DAF stage and after flowering, while the *sldella* and *sltap3* mutants were not. Similar trends were observed in the changes of active GAs throughout the three mutants (Figure 2). The results indicate that the causality of parthenocarpy observed in the *sliaa9* mutant differed from the *sldella* and *sltap3* mutants. Mariotti et al. (2011) discovered that the level of IAA improved after pollination for 2 days in WT [50]. With regard to the relationships between active GA accumulation and parthenocarpy, García-Hurtado et al. (2012) discovered that the overexpression of the citrus GA biosynthetic gene *GA20-oxidase 1* (*CcGA20ox1*) in tomato cv. Micro-Tom stimulated parthenocarpic fruit growth related to increased GA4 content [51]. In this research, *SlGA20ox1* was upregulated in the three mutants, especially the *sliaa9* mutant with pollination (Appendix A); moreover, GA1 and GA4 and precursor GA20 increased from the flowering stage (Figure 2). Therefore, it was assumed that high active GA1, GA4 and precursor GA20 accumulation in an early developmental stage is crucial for fruit set and induced fruit expansion in parthenocarpic tomato, especially the *sliaa9* mutant with pollination. It was posited that *SlIAA9* influences GA biosynthesis genes leading to cell division and multiplication. Additionally, earlier studies demonstrated that the use of artificial plant hormones as the exogenous hormones improved parthenocarpy. For example, 2,4-dichlorophenoxyacetic acid (2,4-D), an artificially produced plant growth regulator with roles comparable with auxin, induced parthenocarpy in pear and stimulated the production of bioactive GA4 fourfold (~36.101 pmol/g FW) compared with pollination (~9.025 pmol/g FW) and induced *PbGA20ox2-like* and *PbGA3ox-1* [52]. It posited that a fruit stimulated by the plant growth regulator accumulates more endogenous hormones compared with a fruit influenced by WT pollination and mutants with pollination.

Some earlier reports demonstrated that parthenocarpy can be caused throughout hormone signalling, and the expression levels of auxin-signalling components AUX/IAAs and ARF family during an early stage in the three mutants with and without pollination were upregulated (Appendix A). *ARF* genes performed critical functions in early fruit development. For instance, *SlARF9* regulates cell division during early tomato fruit development [53]. *ARF3* regulates developmental timing, modelling and gynecium development in Arabidopsis [54]. In this study, the three mutants after flowering induced the expression of *SlARF9* and *SlARF3* (Appendix A). Zhang et al. (2021) found that not only parthenocarpic tomato ‘R35-P’ but also non-parthenocarpic tomato ‘R35-N’ exhibited the increase in *IAA* content after 3DAF [55], suggesting that IAA was independent of pollination. In this study, *SlIAA3*, *SlIAA16*, *SlIAA17* and *SlIAA19* were upregulated in three mutants. In contrast, the auxin biosynthesis genes *SlTAR1*, *ToFZY* and *ToFZY5* were not induced. It also suggests that parthenocarpy caused by *SlIAA9* deficiency may be due to activation of the auxin signalling pathway.

### 4.2. Photosynthesis, Sugar Metabolism and Cell Wall Pathways Are Potential Candidates Contributing Parthenocarpy in the Mutant Tomato

Earlier studies suggested that fruit photosynthesis performs an essential role in fruit establishment, and the chloroplast numbers and cell size conformed to the induction of genes related to photosynthesis and chloroplast biogenesis. The GUN family proteins are localized in the chloroplast. Among them, *GUN4* was upregulated, a regulator of Mg-chelatase activity in the chlorophyll biosynthesis pathway [56]. Moreover, NADPH: protochlorophyllide oxidoreductase catalyses the light-dependent reduction of protochlorophyllide in chlorophyll biosynthesis [57]. In this study, the three mutants demonstrated significant improvements in the expression of *SlNADPH*, *SlPSBS* and *SlGUN4* in ovaries after flowering (Figure 6). Furthermore, our statistical assessment integrating network analysis discovered the activation of photosynthesis in the mutants (M3) (Figure 4). Tang et al. (2015) reported that photosynthesis-associated genes were intensely induced during fruit set [58]. These results reinforced the examination of chloroplast development in parthenocarpy tomatoes and the genes required during fruit setting and transition to mature growth in the mutants. However, chlorophylls, carotenoids and flavonoids declined after flowering (Appendix A). The decline of these secondary metabolite contents at an early stage of development is likely to be due to fruit cell division and multiplication due to energy flow; thus, the accumulation of these metabolites demonstrated a slower rate relative to fruit growth.

In sugar metabolism-related genes/metabolites, the genes encoding carbohydrate absorption and sugar metabolism enzymes demonstrated increased transcripts in ovaries, e.g., *SlSUT1*, *SlFRK2* and *SlTOMSSF* (Figure 6). To produce energy, sugars developed from assimilation are hydrolysed through the glycolysis and TCA cycle. WT pollination and MT with and without pollination produced fructose and glucose after flowering, while high production of sucrose was found in the only flowering date and decreased after flowering (Figure 7). It suggests that the flowering stage requires sucrose as an energy source for fruit setting. *SlHB15A* knockout stimulated parthenocarpy and significantly contributed to rapid ovary growth. A kinetic model of sucrose metabolism projected that the sucrose cycle had increased activity levels in unpollinated ovaries, while it was shut down when sugars rapidly accumulated in vacuoles in fruit-setting ovaries [45]. Moreover, glucose-6-phosphate and fructose-6-phosphate were produced after flowering in the pollinated WT and the mutants (Figure 7). The sugar metabolism-related genes that demonstrated differential expression between the mutants and the WT displayed upregulation during the fruit set developmental process, indicating that the activation of sugar metabolism was an essential process coupled with the induction of pollination-independent fruit set.

WCNA of transcriptome and metabolome data demonstrated that the amounts of cell cycle and cell-wall-related genes were well associated in the three mutants. It suggests that cell division and multiplication were critical in both types of fruit set during the flowering-to-after-flowering transition. Several cell divisions, protein biosynthesis and cell-wall-related genes were upregulated after flowering in the pollinated WT, while their activation transpired earlier in the mutants at the 0DAF stage (Figure 6). Two key regulators, *CDK*s and *cyclin*s, regulate the call cycle. Fourteen cell cycle genes are positively associated with cell multiplication during apple fruit set and development [59]. This study demonstrated that cyclin A2, B-type cyclin, cyclin B1 and cyclin B2 were developed in the mutant tomato (Appendix A), suggesting that cyclins are crucial cell cycle regulators for cell division in the three parthenocarpic mutant tomatoes. Xyloglucan endotransglucosylase/hydrolase (XTH) enzymes plays critical roles in stimulating cell expansion by disassembling xyloglucan [60], and the expression of XTHs (XTH3, XTH9) and endo-xyloglucan transferase (ext) was significantly upregulated in mutant tomato ovaries and pollinated ovaries (Appendix A). Unlike endo-1,4-beta-D-glucanase, expansins promote the long-term extension of extracted cell walls and expansin ability to stimulate rapid cell expansion [61,62]. In this investigation, *expansin precursor 5* (*SlExpa5*) was upregulated in all mutants through development (Figure 6). The high hub association for metabolites included gluconic acid, glucuronic acid and galacturonic acid (Figure 7). These metabolites were linked to the pentose phosphate pathway that was quantitatively a minor route of glucose metabolism and UDP-glucuronic acid and a nucleotide sugar that was a precursor of the cell wall [63]. These outcomes confirm that different pathways are activated in pollination and parthenocarpy tomato in the early developmental stage.

### 4.3. The Proof-of-Concept Approach Could Objectively Extract the Common Genes with Similar Behaviour in Parthenocarpic Tomato

In the MapMan analysis, the expression levels of photosynthesis-linked genes were upregulated in the mutants’ parthenocarpic ovaries, while Severianin ovaries and pollinated ovaries demonstrated similar outcomes (Appendix A). In genes related to photosynthesis, one of the *SlNADPH* cording genes (Solyc12g013710) was commonly upregulated after 0DAF in the three mutants and Severianin without pollination. *SlNADPH* catalyzes the photoreduction of protochlorophyllide to chlorophyllide in higher plants [64]. The *SlNADPH* are negatively regulated by zinc finger transcription factor *SlZFP2* during fruit set [65]. These suggest that the *SlNADPH* may have a crucial role for parthenocarpy.

*SlFRK* repression resulted in a decline in flower development and fruit set in tomato, revealing its crucial role in fruit development [66]. This gene was commonly upregulated in the parthenocarpic tomato used in the study with and without pollination (Figure 6; Appendix A). Shinozuki et al. (2020) discovered that the silencing of *SlFRK2* repressed ovary growth during fruit set in the *sldella* mutant [45], suggesting that the multi-omics approach combined with WCNA can objectively provide candidate genes in sugar metabolism for parthenocarpy.

We found that the significant changes contributed to cell expansion (Figure 6). Among them, *expansin 2* (Solyc06g049050) tended to be upregulated in the mutant and Severianin after 2DAF with and without pollination. Expansins have been regarded as the major wall-loosening constraint causing the extension of plant cell walls. RNA-seq analysis of the suppressed *SlARF5* tomato, which induces parthenocarpy, revealed the significant change of *expansin 2* [67]. These findings suggest that there is probably a tight correlation between cell expansion and parthenocarpy.

## 5. Conclusions

In this study, we conducted transcriptomic, hormonomic and metabolomic analyses to reveal candidate genes and metabolites that may contribute to parthenocarpy using the three well-studied mutants, i.e., tomato lacking *SlIAA9*, *SlDELLA* and *SlTAP3*. Specifically, the resultant candidate genes were sought in the four modules obtained by WGCNA of a parthenocarpic cultivar, ‘Severianin.’ As a result, several genes in the modules related to photosynthesis, sugar metabolism and cell division were commonly observed in the networks of the mutants as well as those of Severianin.

The integrated analysis of multi-omics and the WCNA of transcriptome and metabolome datasets enabled us to reveal additional information necessary for biological activities relevant to parthenocarpy from the tomato mutants with Severianin. Integrated omics analyses emphasised numerous crucial genes and candidate metabolites linked to photosynthesis, sugar metabolism and cell division in parthenocarpic tomato fruit. Understanding the regulation mechanisms of significant genes and metabolites provides us with great potential for subsequent breeding strategies in generating new parthenocarpic tomatoes.

## Figures and Tables

**Figure 1 cells-11-01420-f001:**
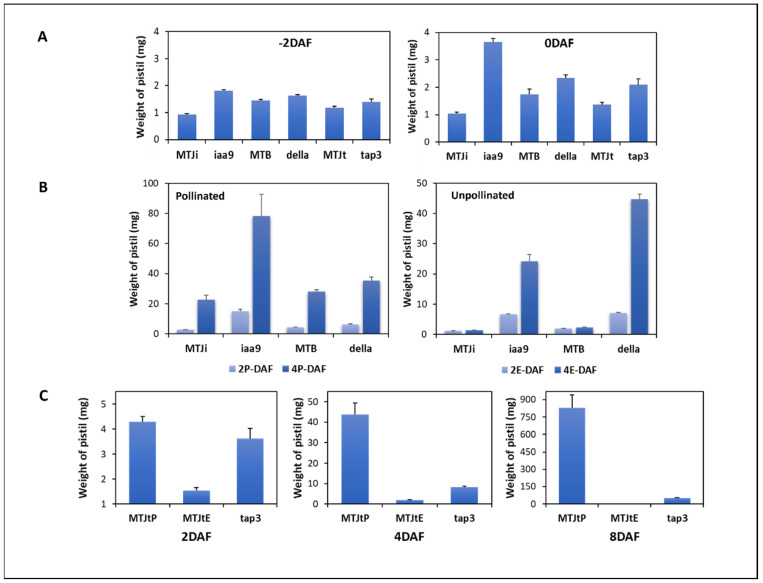
The ovary weight of the three mutants (MT) and the corresponding wild type (WT). (**A**) MT and WT before flowering (−2DAF) and flowering date (0DAF); (**B**) the *sliaa9* and *sldella* mutants and the corresponding WT after flowering 2, 4 DAF; and (**C**), the *sltap3* mutant and MTJt after flowering 2, 4, and 8 DAF. ‘P’ represents pollinated, while ‘E’ represents emasculated. The number of biological replicates, *n* = 6.

**Figure 2 cells-11-01420-f002:**
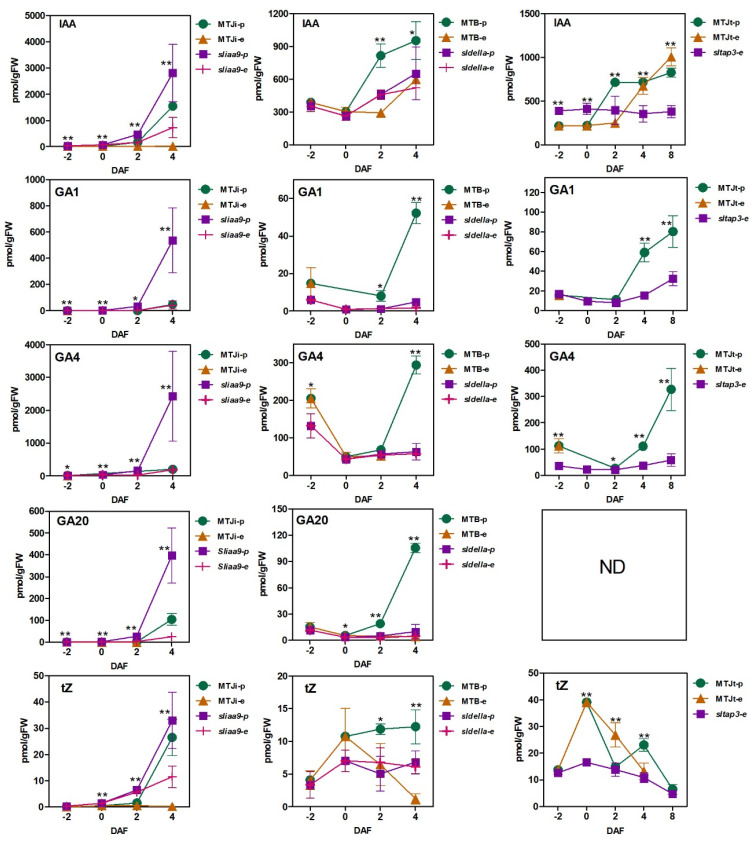
Quantitative plant hormone accumulation during fruit set in MT and WT with and without pollination. IAA, indole-3-acetic acid; GA1, gibberellin 1; GA4, Gibberellin 4; GA20, gibberellin 20; tZ, *trans*-zeatin. ND, not detected. Number of biological replicates, *n* = 3. Significant, * False discovery rate (FDR) < 0.05, ** FDR < 0.005.

**Figure 3 cells-11-01420-f003:**
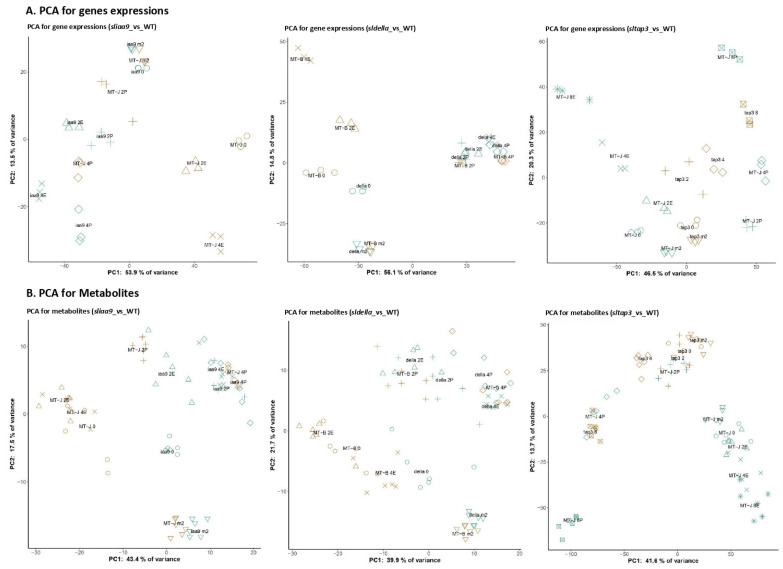
The PCA score scatter plots of transcript and metabolite profiles during early fruit development. The PCA of the transcript levels (**A**) and metabolite (**B**) between MTJi/sliaa9 (MTJ, orange; sliaa9, green), MTB/sldella (MTB, orange; sldella, green) and MTJ/sltap3 (MTJ, green; sltap3, orange) at all four growth stages are shown. Number of biological replicates, *n* = 3 for transcriptmic data; *n* = 5–6 for metabolomic data. Inverted triangle, m2; circle, 0; triangle, 2E; the cross mark, 4E; asterisk, 8E; plus, 2P; rhombus, 4P; and ⊠, 8P.

**Figure 4 cells-11-01420-f004:**
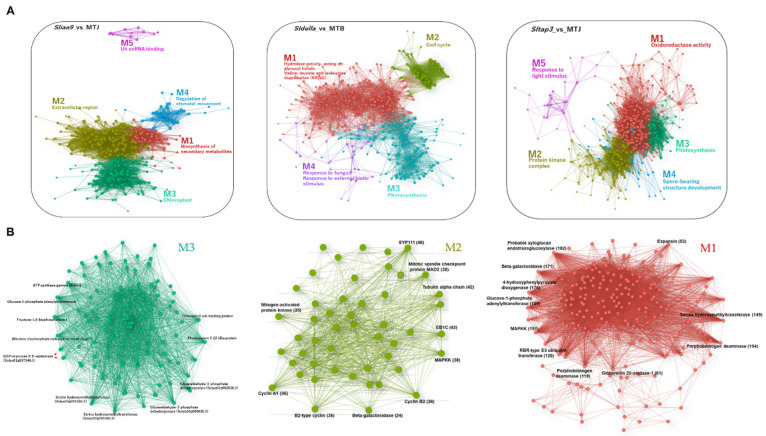
Gene regulatory networks of the tomato parthenocarpic mutants. Weighted gene correlation network analysis (WGCNA) of clustered genes (**A**). Genes were chosen in the matrix consisting of the top 600 genes and metabolites with high values of mean absolute deviation (MAD). Nodes and edges represent genes and co-expression patterns between genes, respectively. The different coloured nodes indicate the five modules evaluated by the fast-greedy modularity optimisation algorithm (M1–M5). Selected GO terms enriched in each module (FDR < 0.05) are shown, and the detailed results of the GO enrichment analysis are shown in Appendix A. Module (**B**) presents the elaborate network structure. Number of biological replicates, *n* = 3.

**Figure 5 cells-11-01420-f005:**
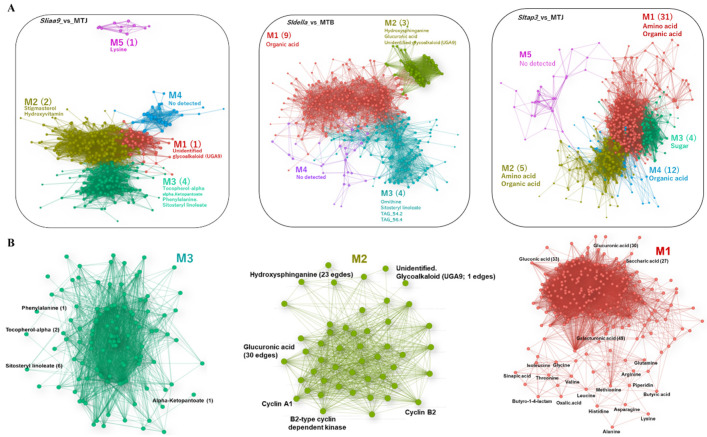
Metabolite correlation networks of the tomato parthenocarpic mutants. The weighted metabolite correlation network analysis (WMCNA) of clustered metabolites (**A**). Metabolites were chosen in the matrix consisting of the top 600 genes and metabolites with high values of mean absolute deviation (MAD). The different coloured nodes indicate the five modules evaluated by the fast-greedy modularity optimisation algorithm (M1–M5). The number and type of metabolites in each module are shown, while detailed outcomes of the metabolites are shown in Appendix A. The elaborate linkage structure of metabolites of each mutant (**B**). Number of biological replicates, *n* = 5–6.

**Figure 6 cells-11-01420-f006:**
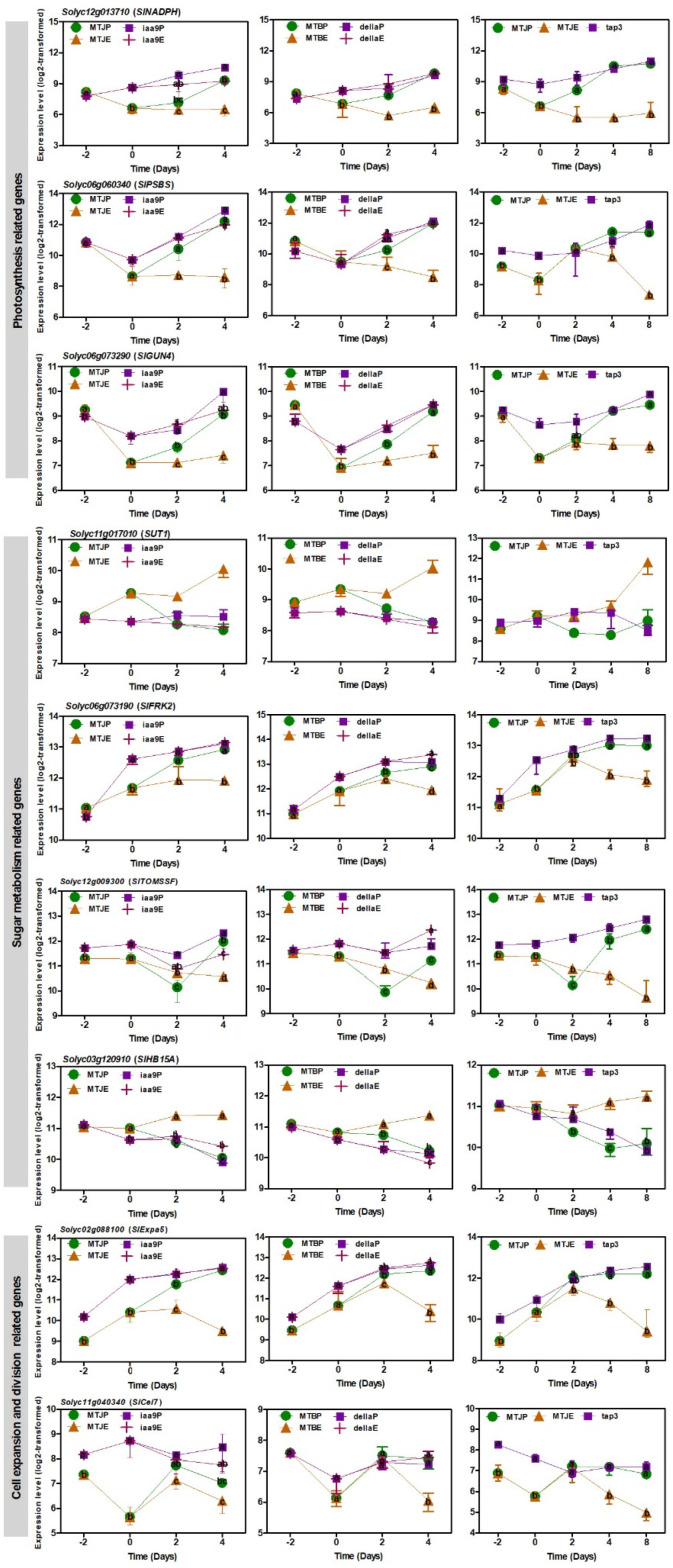
Common genes related to photosynthesis, sugar metabolism, cell division and expansion during fruit set in MT and WT with and without pollination. *SlNADPH*, *protochlorophyllide oxidoreductase 1*; *SlPSBS*, *photosystem II 22 kDa protein*; *SLGUN4*, *tetrapyrrole-binding protein*; *SlSUT1*, *sucrose transporter*; *SlFRK2*, *fructokinase-2*; *SlTOMSSF*, *fruit sucrose synthase*; *SlHB15A*, *SlHOMEOBOX 15A*; *SlExpa5*, *expansin 5*; *SlCel7*, *endo-1,4-beta-D-glucanase*. Number of biological replicates, *n* = 3.

**Figure 7 cells-11-01420-f007:**
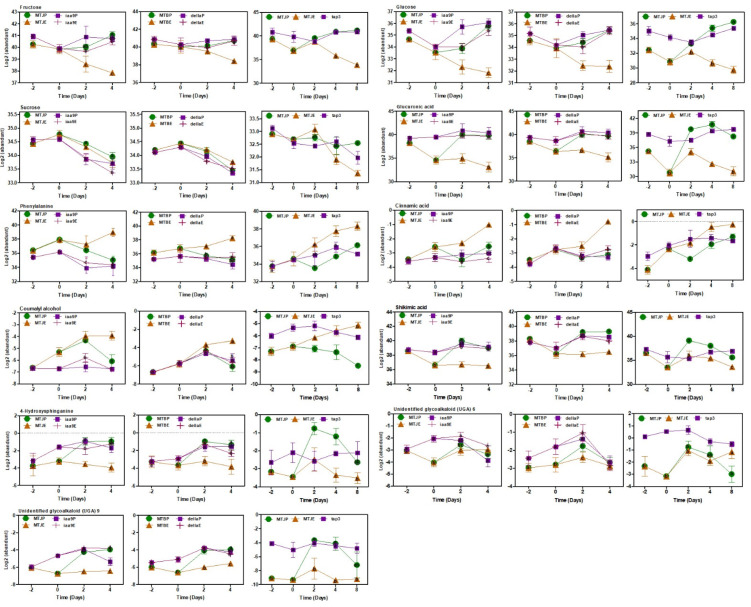
Metabolite accumulation patterns changed in photosynthesis, sugar metabolism and cell division and expansion during fruit set in MT and WT with and without pollination. Number of biological replicates, *n* = 5–6.

**Figure 8 cells-11-01420-f008:**
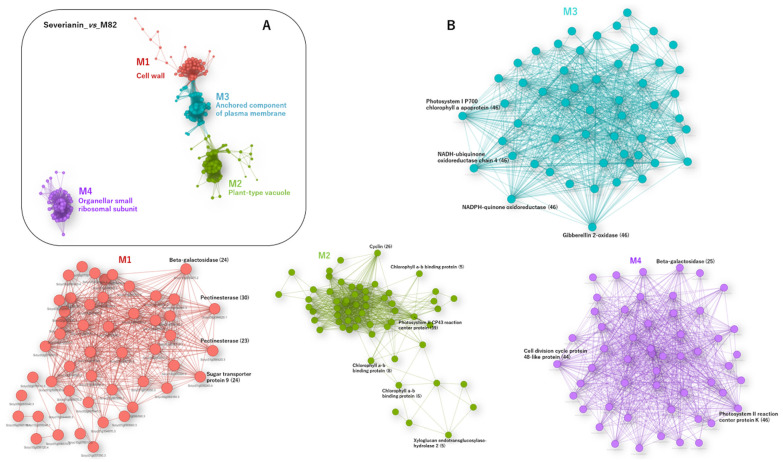
Gene regulatory networks of the parthenocarpic cultivar ‘Severianin’. The WGCNA of clustered genes (**A**). Nodes and edges represent genes and co-expression configurations between genes, respectively. The different coloured nodes show modules assassed by the fast-greedy modularity optimisation algorithm (M1–M4). Selected GO terms enriched in each module (FDR < 0.05) are represented, and the detailed results of the GO enrichment analysis are presented in Appendix A. Module (**B**) presents the detailed network structure in M1, M2 and M4. Number of replicates, *n* = 3.

## Data Availability

Microarray datasets used in the study were deposited to NCBI GEO (accession GSE179122). All short-read data are available for download at the DDBJ Sequence Read Archive under accession number DRA012573. All data are included in the article and Appendix A.

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
