# Peer review of "Transcriptomic, Hormonomic and Metabolomic Analyses Highlighted the Common Modules Related to Photosynthesis, Sugar Metabolism and Cell Division in Parthenocarpic Tomato Fruits during Early Fruit Set"

_cells, 2022, doi:10.3390/cells11091420_

Round 1

Reviewer 1 Report

1. Line 237, “eefficiently” is spelling mistake;
2.   The MS tried to elucidate  some factors in parthenocarpic tomato fruits for early fruit set using mutants, but the results were mainly focused on Transcript and Metabolite Profiles, which were not offer enough data to show the core content.

Author Response

1. Line 237, “eefficiently” is spelling mistake;

(Response) Thank you for your suggestion. We revised the mistake in the revised manuscript with track changes (Line 256). We also corrected other grammatical mistakes carefully in the revised manuscript.

2. The MS tried to elucidate some factors in parthenocarpic tomato fruits for early fruit set using mutants, but the results were mainly focused on Transcript and Metabolite Profiles, which were not offer enough data to show the core content.

(Response) Thank you for your valuable comments. As the reviewer suggested, the current version of the manuscript contains exaggerated descriptions. According to the reviewer’s comments, we revised title and the main text.

Title:

We could extract the common modules throughout the three mutants and a well-known cultivar ‘Severianin’, these showing parthenocarpy. But as in the reviewer’s comments, we could not narrow down the important factors for parthenocarpy during the early fruit set of tomato. So, we changed the title as “Transcriptomic, hormonomic and metabolomic analyses highlighted the common modules related to photosynthesis, sugar metabolism and cell division in parthenocarpic tomato fruits during early fruit set.”

Main text:

We mainly revised Discussion part to remove exaggerated descriptions. We also removed References that are not entirely unnecessary in the manuscript.

Reviewer 2 Report

The manuscript "Cells-1654642-peer-review-v1" is a very important topic and informative, it has new knowledge.

A few points can be corrected by authors:

- The authors should move the part of “Plant materials and growth conditions” from Supplementary data to the Materials and Methods section of the manuscript.

Results

- The quality of figure 1 is Not good, so, please try to rearrange it, and improve the contrast

Conclusions

- The conclusions are very short, compacted, and cannot give meaning, therefore, the authors should rewrite them to summarize the results and reflect the manuscript objectively.

Author Response

The manuscript "Cells-1654642-peer-review-v1" is a very important topic and informative, it has new knowledge.

(Response) Thank you for the positive comment from the reviewer.

A few points can be corrected by authors:

- The authors should move the part of “Plant materials and growth conditions” from Supplementary data to the Materials and Methods section of the manuscript.

(Response) Thank you for the suggestion. We added the phrases about cultivation information to “Plant materials and growth conditions” section from Methods S1 (line 138-157).

Results

- The quality of figure 1 is Not good, so, please try to rearrangeit, and improve the contrast

Conclusions

(Response) Done.

- The conclusions are very short, compacted, and cannot give meaning, therefore, the authors should rewrite them to summarize the results and reflect the manuscript objectively.

(Response) According to the reviewer’s comment, we added phrases to describe summary of the results (line 633-639).

Reviewer 3 Report

Title:
Title nee to be very specific - "Multi-omics analysis highlighted critical factors" - better to highlight any of these factors in the title. At present, the title looks like a fishing expedition study rather than hypothesis-driven research or study with any conclusive finding.  Readers will not be attracted to read "critical factors" having a role in three different physiological processes like "photosynthesis, sugar metabolism and cell division"

Introduction
How relevant it is to say multi-omics when there is the use of Transcriptomic and metabolomics only. Secondly the metabolomics performed in the present study is not extensive enough. 

Materials and methods
Authors need to provide details of biological replication considered for the transcriptomic analysis

Similarly, details of biological replication should be provided in section - "Metabolomic Profiling Analysis"

Results
Figure 1 is very confusing, also need to better arrange the graphs. 

"In the 27 hormones that were assessed " - not sure what authors mean by 27 hormones. 

Conclusion
No doubt the study has used extensive data to understand the molecular mechanism involved in fruit development. Overall the paper is very descriptive but I can understand this need to be descriptive. However, the Authors need to well elaborate on the conclusion section and provide pointwise crispy and clear conclusions. Statements like - this enhanced our understanding or highlighted critical factors will not help to readers. 

Author Response

Title:

Title nee to be very specific - "Multi-omics analysis highlighted critical factors" - better to highlight any of these factors in the title. At present, the title looks like a fishing expedition study rather than hypothesis-driven research or study with any conclusive finding. Readers will not be attracted to read "critical factors" having a role in three different physiological processes like "photosynthesis, sugar metabolism and cell division"

(Response) Thank you for the suggestion. According to the reviewer’s suggestion, we changed the title, “Transcriptomic, hormonomic and metabolomic analyses high-lighted the common modules related to photosynthesis, sugar metabolism and cell division in parthenocarpic tomato fruits during early fruit set” to avoid the exaggerated presentation.

Introduction

How relevant it is to say multi-omics when there is the use of Transcriptomic and metabolomics only. Secondly the metabolomics performed in the present study is not extensive enough.

(Response)

Yes, we conducted transcriptomic and metabolomic analyzes, not including proteomics, ionomics, etc. We thus tried explaining what kinds of omics alayses we used in the study.

About the metabolomics analysis. Using the GC-TOF-MS, UPLC-q-TOF-MS and UPLC- tandem quadrupole-MS, we could detect 406 annotated metabolites. Furthermore, 42 hormones and 6 carotenoids were quantified (Dataset S1, S2). Annotation was carefully done according to the metabolomics community proposed five levels of compound annotations. We selected metabolites that have annotation confidence level 1 and 2 (Level 1: annotations correspond to “identified metabolites” using reference standard match or full 2D structure elucidation, with at least two orthogonal techniques (e.g., retention time (index) and accurate mass for MS-based metabolomics) defining 2D structure confidently, while Level 2: annotations correspond to “putatively identified” compounds, describing a probable structure that is matched to literature data or databases by diagnostic evidence. We think our metabolomic data and hormone analysis is sufficient to conduct WMCNA.

Materials and methods

Authors need to provide details of biological replication considered for the transcriptomic analysis

Similarly, details of biological replication should be provided insection - "Metabolomic Profiling Analysis"

(Response) Thank you for your suggestion. We used three biological replicates for transcriptomic and hormonomic analyses, while 5-6 replicates were used for metabolomic analysis. We added the information in Material and Method section (line 183-185) as well as each Figure legend related to transcriptomic, hormonomic and metabolomic analyses.

Results

Figure 1 is very confusing, also need to better arrange the graphs.

(Response) Done.

"In the 27 hormones that were assessed " - not sure what authors mean by 27 hormones.

(Response)

Thank you for your suggestion. This is our mistake. We quantified 42 hormones including IAAs, GAs, CKs, ABA, SA and JA (Dataset S1). We corrected the number and add hormones’ names (line 251-253).

Conclusion

No doubt the study has used extensive data to understand the molecular mechanism involved in fruit development. Overall the paper is very descriptive but I can understand this need to be descriptive. However, the Authors need to well elaborate on the conclusion section and provide pointwise crispy and clear conclusions. Statements like - this enhanced our understanding or highlighted critical factors will not help to readers.

(Response) As in the correction of the title, we excluded the phrase “critical factors” in the revised manuscript. Instead, we emphasized that the three modules were commonly observed in the networks of the three mutants as well as Severianin (line 633-639).   

Round 2

Reviewer 1 Report

The revised Ms meets the requirement for publish.

Reviewer 3 Report

Kusano et al. have addressed all of my concerns and considered the typos and edits. The improved version looks appropriate.